# Vaccine-Induced T-Cell and Antibody Responses at 12 Months after Full Vaccination Differ with Respect to Omicron Recognition

**DOI:** 10.3390/vaccines10091563

**Published:** 2022-09-19

**Authors:** Franz Mai, Johann Volzke, Emil C. Reisinger, Brigitte Müller-Hilke

**Affiliations:** 1Core Facility for Cell Sorting and Cell Analysis, Rostock University Medical Center, 18055 Rostock, Germany; 2Division of Tropical Medicine and Infectious Diseases, Center of Internal Medicine II, Rostock University Medical Center, 18055 Rostock, Germany

**Keywords:** SARS-CoV-2, COVID-19, homologous and heterologous vaccination regime, B cell memory, T cell memory, CD4, CD8, antibodies, Wuhan-Hu-1 wild-type, Omicron variant of concern

## Abstract

More than a year after the first vaccines against the novel SARS-CoV-2 were approved, many questions still remain about the long-term protection conferred by each vaccine. How long the effect lasts, how effective it is against variants of concern and whether further vaccinations will confer additional benefits remain part of current and future research. For this purpose, we examined 182 health care employees—some of them with previous SARS-CoV-2 infection—12 months after different primary immunizations. To assess antibody responses, we performed an electrochemiluminescence assay (ECLIA) to determine anti-spike IgGs, followed by a surrogate virus neutralization assay against Wuhan-Hu-1 and B.1.1.529/BA.1 (Omicron). T cell response against wild-type and the Omicron variants of concern were assessed via interferon-gamma ELISpot assays and T-cell surface and intracellular cytokine staining. In summary, our results show that after the third vaccination with an mRNA vaccine, differences in antibody quantity and functionality observed after different primary immunizations were equalized. As for the T cell response, we were able to demonstrate a memory function for CD4+ and CD8+ T cells alike. Importantly, both T and antibody responses against wild-type and omicron differed significantly; however, antibody and T cell responses did not correlate with each other and, thus, may contribute differentially to immunity.

## 1. Introduction

Approximately one year after the December 2019 outbreak of novel SARS-CoV-2, the first vaccines against COVID-19 had been approved. Successfully established vector-based manufacturing methods leading to AstraZeneca’s Vaxzevria (AZD1222) were complemented with novel methods using mRNA technology, leading to Pfizer-BioNTech’s Comirnaty (BNT162b2) and Moderna’s Spikevax (mRNA-1273). Although the manufacturing methods of these vaccines differ, they all result in translation of the original spike protein of SARS-CoV-2, eliciting both B and T cell responses. In phase 3 trials, the vaccines showed efficacies of 95% after two doses of BNT162b2 [1] and 70% after two doses of AZD1222 [2] against severe COVID-19 cases, respectively. However, the durability of immune protection, the impact of homologous vs. heterologous prime boost immunization vs. SARS-CoV-2 exposure, as well as protection from variants of concern, are still a matter of debate and need to be constantly evaluated as we move through this pandemic.

While a two-dose vaccination with any of the above mentioned vaccines elicited strong antibody responses even in low responders, we and others were able to demonstrate that a heterologous prime boost regimen combining AZD1222 with any of the mRNA vaccines not only elicited a significantly higher number of antibodies, but also improved neutralizing capacity [3,4]. On the other hand, spike protein specific antibodies decreased over time with an estimated half-life of approximately 55 days and this decline in humoral protection seemed independent of the vaccine [5,6]. However, the impact of a third and possibly fourth vaccination on durability are still obscure as is the contact with the virus itself, leading to hybrid immunity [7]. As for the cellular response to vaccination, immunity provided by the adenoviral vector-based AZD1222 was shown to be somewhat lower and to wane faster compared to the mRNA vaccines [8]. However, half-lives of memory CD4- and CD8-positive T cells were reported to range between 4 and 6 months and, thus, spike-specific T cells seemed to be more robust than antibodies [9,10,11]. As more variants of the original Wu-Hu-1 SARS-CoV-2 strain emerged, carrying up to 36 mutations in the recent B.1.1.529/BA.1-5 (Omicron) variant, break-through infections increased and questioned the efficacy of vaccination [12,13]. Indeed, the neutralizing capacities of spike-specific antibodies against variants of concern declined and this was true not only for delta but also for Omicron [4,14]. However, T cells play a major role in mitigating disease progression and fighting infection [15], and the conserved recognition of variants by memory CD8 T cells possibly contributes to the confinement of severe courses following infection with Omicron [16,17,18]. As of yet, questions remain about how the vaccine, the vaccination regimen, or exposure to SARS-CoV-2 itself impact on immune memory over time [19]. To test for potential differential efficacy resulting from different primary immunizations, here, we set out to collect longitudinal data on the durability of B and T memory cells 12 months after full vaccination and a boost by either vaccination, infection, or both.

## 2. Materials and Methods

### 2.1. Blood Samples

Human blood samples were collected via venipuncture at 12 months after primary immunization. EDTA blood was centrifuged at 1500× *g* for 10 min to obtain plasma and untreated blood was centrifuged at 2000× *g* for 10 min to obtain serum. Both were frozen at −80 °C for subsequent determination of IgG antibodies towards spike-RBD and nucleocapsid as well as for neutralization assays. Peripheral blood mononuclear cells (PBMCs) were isolated by density gradient centrifugation (Ficoll-PaqueTM PLUS, Cytiva, Marlborough, MA, USA) and stored in heat-inactivated fetal calf serum (FCS, Thermo Fisher, Waltham, MA, USA) containing 10% dimethyl sulfoxide (Sigma-Aldrich, St. Louis, MO, USA) at −80 °C for later use in Interferon Gamma assay (ELISpot) and T cell analysis by flow cytometry (Cytek Biosciences, Fremont, CA, USA).

### 2.2. T Cell Activation Marker and Cytokine Staining after Re-Stimulation with either BNT162b2 or Wuhan-Hu-1 and Omicron Peptide Pools

PBMCs were thawed and transferred to RPMI cell culture medium containing 1 mM pyruvate, 2 mM L-glutamine, 10 mM HEPES, 10% FCS and 100 U penicillin/0.1 mg streptomycin (PAN-Biotech, Aidenbach, Germany). After centrifugation at 400× *g* for 5 min and resuspension in RPMI cell culture medium, cell count was performed by Cytek^®^ Aurora using SpectroFlo software version 2.2.0.3 (Cytek Biosciences, Fremont, CA, USA). Live/dead differentiation was conducted in 4′,6-diamidino-2-phenylindole (DAPI at a dilution of 1:720,000; Biolegend, San Diego, CA, USA). Four samples of 8 × 10^5^ PBMCs each were seeded in duplicates into 96-well U-bottom cell culture plates. One sample remained unstimulated throughout the experiment and served as a negative control. The second sample was stimulated with 1 µg of the vaccine BNT162b2 (Pfizer-BioNTech, Mainz, Germany). The remaining two samples were left untreated for 18 h before the addition of 0.2 μg of peptide pools representing either the Omicron variant (PepTivator^®^SARS-CoV-2 Prot_S B.1.1.529/BA.1 Mutation Pool) or the Wu-Hu-1 wild-type strain (PepTivator^®^SARS-CoV-2 Prot_S B.1.1.529/BA.1 WT Reference Pool) (Miltenyi Biotec, Bergisch Gladbach, Germany). After a total of 20 h of incubation, 1 µg of brefeldin A (Sigma-Aldrich, St. Louis, MO, USA) was added to all wells. After a total of 24 h, cells were harvested, centrifuged at 400× *g* for 5 min, and duplicates were pooled in autoMACS Running Buffer (Miltenyi Biotec, Bergisch Gladbach, Germany). Subsequently, cells were washed with PBS, centrifuged for 5 min at 400× *g* and incubated with 1:2000 diluted Zombie NIR (BioLegend, San Diego, CA, USA) for 20 min at room temperature in darkness. Thereafter, non-specific binding sites were blocked with 2.5 µL of FCS (Fisher Scientific, Pittsburgh, PA, USA), 1.25 µL of True-Stain MonocyteTM Blocker, and 1.25 µL of Human TruStain FcXTM (BioLegend, San Diego, CA, USA) and incubated for 15 min on ice in darkness. The following antibody:fluorophore combinations were used for surface staining: 2.5 µL of CD3:FITC (clone UCHT1), 0.625 µL of CD4:BV750 (SK3), CD8:BV570 (RPA-T8) (BioLegend, San Diego, CA, USA), and 1.25 µL of CD137:BV480 (4B4-1) (BD Biosciences, Franklin Lakes, NJ, USA). After incubation on ice for 15 min in darkness, samples were fixed with fixation buffer (BD Bioscience, Franklin Lakes, NJ, USA) for 20 min at room temperature in darkness, followed by three washes with intracellular staining and permeabilization wash buffer (Invitrogen, Carlsbad, CA, USA). Subsequently, non-specific binding sites were again blocked as described above for 15 min at room temperature in darkness. For intracellular cytokine staining, the following antibody:fluorophore combinations were used: 2.5 µL each of IFNγ:PerCP/Cy5. 5 (4S.B3), IL-2:BV650 (MQ1-17H12) (BioLegend, San Diego, CA, USA), IL-4:PE/Dazzle594 (MP4-25D2), IL-10:BV421 (JES3-907), and TNFα:PE/Cy7 (Mab11). After 30 min of incubation at room temperature in darkness, a final wash with intracellular staining and permeabilization wash buffer was performed before samples were measured by flow cytometry using Cytek Aurora with SpectroFlo software version 2.2.0.3.

### 2.3. Measurement of IgG against SARS-CoV-2 Spike Protein and SARS-CoV-2 Nucleocapsid

Electrochemiluminescence immunoassay (ECLIA) (Elecsys^®^ Anti-SARS-CoV-2 S, Roche Diagnostics, Mannheim, Germany) was performed according to the manufacturer’s instructions. Frozen patient serum was thawed and incubated with RBD (spike or nucleocapsid) antibodies, followed by the addition of dye antibodies. Final measurements were performed on Cobas E411 (Roche Diagnostics, Rotkreuz, Switzerland). Measured U/mL correlated strongly with the international WHO standard BAU/mL (U = 0.972 * BAU; Pearson r = 0.99996).

### 2.4. Neutralizing Capacity against Wuhan-Hu-1 and B.1.1.529/BA.1 (Omicron)

The SARS-CoV-2 Surrogate Virus Neutralization Test (sVNT) kit (GenScript, Piscataway, NJ, USA) was used as the neutralization assay and was performed according to the manufacturer’s instructions. Initially, frozen patient plasma was thawed, then centrifuged at 10,000× *g* for 5 min to remove precipitates, followed by a 1:10 dilution. HRP peptides Wuhan-Hu-1 (SARS-CoV-2 spike protein (RBD, Avi and His Tag)-HRP) and Omicron (SARS-CoV-2 spike protein RBD-HRP, Omicron Variant, His Tag) (GenScript) were diluted at 1:1000. Diluted plasma samples and diluted HRP peptide samples were mixed in equal parts. After 30 min of incubation at 37 °C, samples were pipetted onto ELISA capture plates and incubated for an additional 15 min at 37 °C. After four washing cycles, substrate solution was added and incubated for 15 min at room temperature before stop solution was added to terminate the reaction. Photometric measurements of the capture plate were performed at 450 nm using the InfiniteM200 (Tecan, Männeheim, Switzerland). Optical densities (OD) were used for calculation: Neutralizing Capacity = (1 − OD_sample_/OD_Neg_._Ctrl_) × 100%.

The first WHO International Reference Panel for anti-SARS-CoV-2 immunoglobulin 20/268 (NIBSC, South Mimms, UK) was used for calibration of neutralizing antibodies to International Units (IU/mL). To that extent, the five WHO serum samples with defined IUs were diluted at 1:10 and mixed at 1:2 with the diluted HRP peptide sample. ODs of these standard samples were fitted to an exponential function (exponential regression), which was then applied to the ODs measured for our vaccine serum samples.

### 2.5. Anti-Human Interferon Gamma ELISpot

Frozen PBMCs were processed as described in 2.2. Three samples of 5 × 10^5^ PBMCs each were seeded into the wells of a V-bottom plate. The first sample remained untreated, the second and third were mixed with 0.2 µg of peptide pools representing either the Wu-Hu-1 wild-type strain (PepTivator^®^SARS-CoV-2 Prot_S B.1.1.529/BA.1 WT Reference Pool) or the Omicron variant of concern (PepTivator^®^SARS-CoV-2 Prot_S B.1.1.529/BA.1 Mutation Pool) (Miltenyi Biotech). Subsequently, all samples were transferred to an anti-interferon gamma coated U-bottom ELISpot plate (Human IFN-gamma ELISpot Kit, R&D Systems, Minneapolis, MN, USA). After 30 min incubation at 37 °C in CO2 incubator (Binder, Tuttlingen, Germany), each well was filled up to 200 µL with RPMI. Thereafter, incubation was continued at 37 °C and 5% CO_2_ for another 23.5 h. After a total incubation time of 24 h, samples were washed four times and IFNγ detection antibodies were added into each well, followed by incubation over night at between 2 and 8 °C. On the next day, four washes were performed again, alkaline phosphatase conjugated streptavidin was added and incubated for 2 h at room temperature in the dark. After another four washes, BCIP/NBT substrate was pipetted into each well, incubated for another hour at room temperature, again in the dark. After a final wash, the plates were left to dry at 37 °C for 30 min. Scanning and counting of the ELISpot plates was performed by ImmunoSpot 5.0 Analyzer with software version 5.0.9.15 (CTL Europe, Bonn, Germany).

### 2.6. Statistics

Data were tested for Gaussian distribution using the Kolmogorov–Smirnov test. Pairwise comparisons of data not following Gaussian distribution were performed via Wilcoxon matched-pairs signed rank or Friedman tests for two and three groups, respectively. Unpaired comparisons of three groups were performed via Kruskal–Wallis followed by Dunn’s multiple comparisons tests. Contingency table analyses were performed via Fisher’s exact (for unpaired data) or McNemar’s test when investigating paired data. Correlation analyses were performed via Spearman rank correlation. Statistical assays were performed with GraphPad InStat^®^ version 3.10 for Windows (GraphPad Software, San Diego, CA, USA) or IBM SPSS Statistics Version 27 (IBM, Armonk, NY, USA). Graphs were created with SigmaPlot 13.0 (Inpixon, Palo Alto, CA, USA) and with BioRender.com (accessed on 11 September 2022), respectively.

### 2.7. Ethic Commitment

This study was approved by the ethics committee of the Rostock University Medical Center under the file number A 2020-0086. Written informed consent was provided by all participants.

## 3. Results

### 3.1. Study Population

Peripheral blood samples from a total of 182 vaccinees were taken approximately one year after primary immunization against SARS-CoV-2 (Figure 1). In detail, 120 vaccinees had been primed with AZD1222 followed by secondary immunization with either AZD1222 (*n* = 57) or BNT162b2 (*n* = 63) after 12 weeks, followed by a boost with either BNT162b2 (*n* = 97) or mRNA-1273 (*n* = 11) 6 months later. The remaining 62 vaccinees had been primary and secondary immunized with BNT162b2 and 57 of these were boosted with an mRNA vaccine 9 months thereafter. For all those boosted, peripheral blood was taken 2 months after the boost. For those who had decided against a boost (*n* = 7) or had not yet been eligible for a boost due to recent SARS-CoV-2 infection (*n* = 2), removal of peripheral blood was scheduled at 11 months after primary immunization with AZD1222 and 12 months after BNT162b2, respectively. A total of 6 vaccinees had tested PCR-positive and anti-nucleocapsid positive within 2 months of blood withdrawal. Another 2 from the AZD1222/BNT162b2 group had been anti-nucleocapsid positive at first blood withdrawal with infections either unknown or before the first immunization, respectively (Figure 1).

### 3.2. Homologous and Heterologous Vaccination Regimen Involving AZD1222 and BNT162b2 Result in Comparable T Cell Memories

In order to investigate whether the various prime boost regimen shaped different T cell memory responses against the spike protein, PBMC from 10 vaccinees of each of the three vaccination groups were re-stimulated in vitro. To address CD4 T helper and CD8 positive cytotoxic T cells alike, we used the original BNT162b2 vaccine for re-stimulation and analyzed activation-induced markers as well as intracellular cytokines analyzed via flow cytometry. The gating scheme is provided in Appendix A. Indeed, re-stimulation produced for CD4 as well as CD8 T cells significantly increased numbers of activation marker CD137- and IL-2- and IFNγ-positive cells (Figure 2). Of note, homologous and heterologous primary/secondary immunizations yielded comparable recall responses, as did infection with the virus. In summary, our data show that (i) the BNT162b2 vaccine is capable of eliciting in vitro T cell recall responses and (ii) homologous and heterologous primary/secondary immunizations involving either AZD1222 or BNT162b2 followed by an mRNA boost or SARS-CoV-2 infection resulted in comparable T cell memories.

### 3.3. Boosting with an mRNA Vaccine Balanced Differences in Antibody Levels That Had Been Apparent at 6 Months after Primary Immunization

Because antibody responses following homologous or heterologous vaccination regimen were significantly different at six months after primary immunization [ 4], we were curious as to whether boosting with an mRNA vaccine would compensate for these differences. Figure 3A shows that at two months after the boost, the median antibody level in the BNT162b2 only vaccination regimen was highest by trend with 12,233 BAU/mL and most of the samples at and above the upper limit of detection (25,000 BAU/mL) were in this group. However, there were no statistical differences between the three vaccination regimen. Of note, hybrid immunity with infections within the last two months led to antibody levels around and above the upper limit of detection. In contrast, infections longer than five months ago resulted in antibody levels in the lower quartiles (Figure 3A). In summary, our data show that boosting with an mRNA vaccine—or infection with SARS-CoV-2—balanced differences in antibody levels that were apparent at six months after primary/secondary immunizations.

### 3.4. Antibody Mediated Neutralizing Capacity of the Omicron Spike Protein Showed a 15- to 80-Fold Reduction Compared to Wu-Hu-1

We next asked whether the antibodies resulting from the three different vaccination regimens differed functionally. To that extent, we performed surrogate virus neutralization assays using the wild-type Wuhan-Hu-1 and the Omicron variants of the spike protein. In order to avoid the assay’s upper limit of detection and obtain neutralization capacities ranging between 0 and 100%, serum samples needed to be diluted between 15- and 80-fold. Figure 3B presents the results; however, statistical comparisons were inappropriate as they would not adequately take the different serum dilutions into account.

To circumvent this dilemma, we turned to the WHO International Reference Panel for anti-SARS-CoV-2 immunoglobulin and transferred the neutralization capacities against WU-Hu-1 into IU/mL. Median IU/mL for the different vaccination groups were 6640 (AZ/AZ), 5482 (AZ/BNT), and 7841 (BNT/BNT), respectively. Comparing the three vaccination groups yielded a significant difference between AZ/BNT and BNT/BNT (Appendix A). Moreover, hybrid immunity for all three vaccination regimens resulted in neutralization capacities, which were predominantly located in the upper quartiles. In contrast, the absence of a boost led to lowest neutralization capacities.

Neutralization capacities against the Omicron variant ranged between 0 and 96.5% without the need for any pre-dilution (Figure 3C). Sera from convalescent individuals again tended to more efficient neutralization capacities while individuals who had not been boosted ranged at the lower end (Figure 3C). As the WHO reference panel does not define IUs for the neutralization of Omicron, an exact quantification as calculated for the neutralization capacities against Wu-Hu-1 could not be conducted for Omicron. Furthermore, because surrogate neutralization assays against Wu-Hu-1 and Omicron resulted in percentages between 0 and 96.5%, yet required between 15- and 80-fold serum dilutions for the Wu-Hu-1 assays, a statistical evaluation of the differences between Wu-Hu-1 and Omicron neutralization was not possible. We also asked whether high concentrations of antibodies against the spike protein would automatically predict high neutralization capacities and, therefore, performed correlation analyses. To that extent, we correlated the BAU/mL of all 182 vaccinees with the respective neutralization capacities against the Wu-Hu-1 spike protein measured in IU/mL and against the Omicron variant, measured in% neutralization. Resulting coefficients were 0.7405 and 0.7433, respectively (Appendix A).

In summary, our data show that (i) following an mRNA boost or infection, serum antibodies efficiently neutralized the Wu-Hu-1 spike protein, independent of the vaccination regimen and (ii) neutralization of the Omicron spike variant was less efficient.

### 3.5. T Cell Memory and Antibody Responses Did Not Correlate with Each Other

Correlating the neutralization capacities against Wu-Hu-1 and Omicron with each other resulted in a coefficient of 0.6859, indicating a moderate connection. We were also intrigued as to whether the magnitude of the antibody response would parallel the T cell response and, therefore, be correlated for the 34 vaccinees who had been screened for their BNT162b2 memory, antibody concentrations with numbers of activated or cytokine positive T cells (Appendix A). However, there were no correlations. These data show that (i) higher antibody concentrations resulted in higher neutralization capacities against wild-type spike protein and its Omicron variant, (ii) higher neutralization capacity against wild-type resulted in higher neutralization against Omicron, however, (iii) higher antibody concentrations did not correlate with elevated T cell memory.

### 3.6. T Cell Memory against the Wu-Hu-1 Spike Protein and Its Omcicron Variant Were Comparable

In order to assess whether the drop in Omicron recognition observed for anti-spike antibodies also applied to the T cell memory, we performed ELISpot assays and re-stimulated 5 × 10^5^ PBMCs with peptide pools representing the Wu-Hu-1 spike protein and its Omicron variant, respectively. Figure 4 segregates the individual results according to the vaccination regimen. The mean plus 2 SEM numbers of IFNγ-producing spots in the absence of stimulating peptides set the threshold, above which responders were defined. McNemar’s tests compared the ratios of responders to non-responders within a given vaccination group and the resulting *p* values of 1.0 (AZ/AZ), 0.1489 (AZ/BNT), and 0.2888 (BNT/BNT) confirm comparable IFNγ responses towards Wu-Hu-1 and Omicron peptide pools in all three groups. In summary, these data show comparable immunological memories against Wu-Hu-1 and Omicron, independent of the vaccination regimen.

However, when comparing the three vaccination regimens for their efficiencies in generating spike-specific T cells, there were significant differences. A Fisher’s exact test comparing the ratios of responders to non-responders against the Wu-Hu-1 peptides between AZ/AZ and BNT/BNT resulted in a raw *p* value of 0.0167, confirming less responders in the AZ/AZ group. Likewise, comparing the three vaccination groups for their ratios of responders and non-responders against the Omicron peptide pool revealed more significant differences. Fisher’s exact tests resulted in *p* values of 0.0076 comparing AZ/AZ to the AZ/BNT and of 0.019 to the BNT/BNT group, respectively (Figure 4). These results imply that homologous priming with AZD1222 yielded significantly less spike-specific T cells.

In order to gain a more detailed insight into the differences between T helper cell responses against the Wu-Hu-1 spike protein and its Omicron variant, we re-stimulated PBMCs from a limited number of randomly chosen samples (AZ/AZ: *n* = 3; AZ/BNT: *n* = 2; BNT/BNT: *n* = 3) with the respective peptide pools and screened for intracellular cytokines. Friedman tests for pairwise comparisons of cytokine positive cells under non-stimulated, Wu-Hu-1 and Omicron stimulated conditions resulted in *p* values of 0.6543 for IL-4, 0.2359 for TNFα, and 0.1495 for IL-10, rendering a specific response unlikely. In contrast, *p* values of 0.0789 for IL-2 and 0.0230 for IFNγ prompted us to look more closely. Figure 5 presents significant increases in cytokine positive cells following re-stimulation with the Wu-Hu-1 spike protein peptide pools. Likewise, re-stimulation with the Omicron peptide pools also yielded increased numbers of cytokine positive cells, even though these differences did not quite reach statistical significance. Re-stimulation with Wu-Hu-1 and Omicron peptide pools did not result in any further significant responses (see Appendix A). In summary, these intracellular cytokine profiles reveal a minor drop in Omicron compared to Wu-Hu-1 recognition for T cells.

## 4. Discussion

The present study pursued three goals: (i) to monitor T and B cell memory at one year after primary immunization, (ii) to compare the T and B cell memory against the SARS-CoV-2 wild-type Wuhan-Hu-1 strain and its Omicron variant, and (iii) to assess whether a different immunization regimen shaped different immune memories.

Serum concentrations of antibodies against the SARS-CoV-2 spike protein increased significantly between 6 and 12 months after primary immunization—from medians of 490 to 9220 BAU/mL for the AZ/AZ group, from 2180 to 8810 BAU/mL for the AZ/BNT group, and from 680 to 12,230 BAU/mL for the BNT/BNT group, respectively [4]. These increases are significant and confirm efficiency of the boost [20]. Moreover, the observation that discrepancies between the vaccination regimen, which were apparent at earlier time points, were compensated for after the boost, shows the superiority of mixing and matching different vaccines. In this context, our data indicate that BNT162b2 and mRNA-1273 achieve comparable outcomes. Assuming that the half-lives of anti-spike antibodies following the boost are comparable to the 55 days determined for the primary/secondary immunization, there will be a significant extension of the humoral memory due to the boost [5]. However, as of yet, both the durability of antibodies after the third vaccination and the protective threshold of antibodies are still under debate.

As for the T cell memory at one year after primary immunization, here, we elicited recall responses via stimulation with the BNT162b2 vaccine as well as with peptide pools representing the spike protein. Both assays were shown to induce CD4 as well as CD8 recall responses [21,22]. Upon BNT162b2 stimulation, we measured significant upregulations of the activation-induced marker CD137 and elevated numbers of IL-2 and IFNγ positive CD4 and CD8 cells alike. These results confirm Th1 immunity, as well as CD8 memory, and indicate successful vaccination, as Th1 responses are crucial for the efficient control of SARS-CoV-2 [23,24]. Indeed, patients with elevated numbers of IFNγ secreting T cells reactive against SARS-CoV-2 proteins were shown to be better protected against the virus [25,26]. While BNT162b2 re-stimulation resulted in comparable T cell responses among the various vaccination regimen, the ELISpot yielded significantly less responders for the AZ/AZ group. These apparently contradictory results may either result from less developed T cell memories following homologous AZD1222 priming [16,27], a more robust vaccine-induced recall response followed by intracellular staining assays on the other [22], or simply different numbers of samples analyzed. However, it also remains a possibility that using peptide pools for the ELISpot assay implies a restriction, as presentation of peptides may be more dependent on HLA haplotypes. Along these lines, another puzzling observation referred to the absence of an increase in positive ELISpots at 12 compared to 6 months [4] for which there may be two possible explanations. For one, we used different peptide pools at 6 and at 12 months in order to allow for the direct comparison of T cell reactivity against Wu-Hu-1 and Omicron at this later time point. These peptides may cover epitopes of lower dominance. For two, T cell responses to vaccination may be more robust than B cell responses, and even though they wane over time, they were suggested to benefit less from booster immunizations and, therefore, may present with less impressive enhancements from 6 to 12 months [8,9,10].

Here, we confirmed a marked decline in antibody neutralization of the Omicron variant compared to the Wuhan-Hu-1 wild-type strain [28]. Even though the surrogate neutralization assay applied did not allow for a direct comparison of neutralization capacities against wild-type and variant proteins, the need for different serum dilutions in order to obtain results in comparable neutralization ranges indicates significant differences.

In contrast, T cell responses against the spike protein as measured in ELISpots, did not show any difference between wild-type and Omicron peptide pools. Likewise, intracellular IL-2 and IFNγ in response to these peptide pools only showed a minor decline in Omicron compared to wild-type. Indeed, T cell responses were previously shown to be unaffected by variants of concern, likely because T cell responses against SARS-CoV-2 are highly multi-antigenic and multi-specific, with many different epitopes being recognized by T helper and cytotoxic T cells in a given individual [16,17,25,28,29,30]. Of note, serum concentrations of antibodies recognizing the spike protein correlated strongly with their capacity to neutralize both, the wild-type spike protein and its Omicron variant. Moreover, the neutralization capacities also correlated strongly with each other. These results confirm the notion that protection from COVID-19 is a direct function of titer [31]. However, we did not find any correlation between antibody concentration or neutralizing capacity and T cell memory. To this latter finding, there are conflicting publications. On the one hand, a subset of CD4+ T helper cells, called follicular helper cells (Tfh), is not only critical for the expansion, affinity maturation, and memory development of B cells, but also for Th1 cells, which foster the development of CD8+ T cell memory [32,33]. It is, therefore, plausible to assume that anti-Spike titers correlate with the spike-specific T cell response. Indeed, that could be confirmed after the second immunization with an mRNA vaccine [34]. On the other hand, however, both the vaccine itself and time seem to impact the correlation between anti-Spike antibody responses and T cell responses. It was, thus, shown that in AZD1222 vaccinees, neither anti-Spike antibodies, memory B, memory CD4+, nor memory CD8+ T cells correlated with each other [27]. Likewise, while T follicular helper cell responses at an early time point after mRNA vaccination correlated with subsequent titers of neutralizing antibodies, a comparable relatedness between memory CD4 or CD8 positive T cells and antibody titers could not be confirmed [27]. Along these lines, a lack of correlation between neutralizing antibodies and disease severity in primary COVID-19, as well as reports of healthy individuals successfully controlling SARS-CoV-2 with little or no neutralizing antibodies, support a role of the cellular independent of the humoral memory [8,35,36].

Since our cohort comprised a mere ten convalescent individuals, here, we only touched the issue of hybrid immunity. Previous reports supported an impressive improvement in antibodies and breadth of neutralization to SARS-CoV-2 variants following infection [8,37]. Our data confirm these findings, yet, in addition, suggest that mixing and matching of vaccines yields comparable results. Importantly, the fact that for eight vaccinees, their previous infection dated back as long as the boost for those being re-immunized with an mRNA vaccine, supports this finding. Moreover, homologous priming with BNT162b2 in our cohort particularly set the stage for maximum anti-spike IgG concentrations and neutralization against Wu-Hu-1 and Omicron. Whether our data support the notion that hybrid immunity is particularly durable is difficult to decide, as in our cohort there were only two individuals whose infection dated back more than 5 months.

There are limitations to our study, and they relate to the small numbers of individuals analyzed. Moreover, analyzing differentiation markers on SARS-CoV-2 responsive T cells will help to characterize the memory compartment in more detail. However, one of the more urgent questions to clarify in the future relates to the durability of T and B cell memory and the protective threshold of antibodies.

## Figures and Tables

**Figure 1 vaccines-10-01563-f001:**
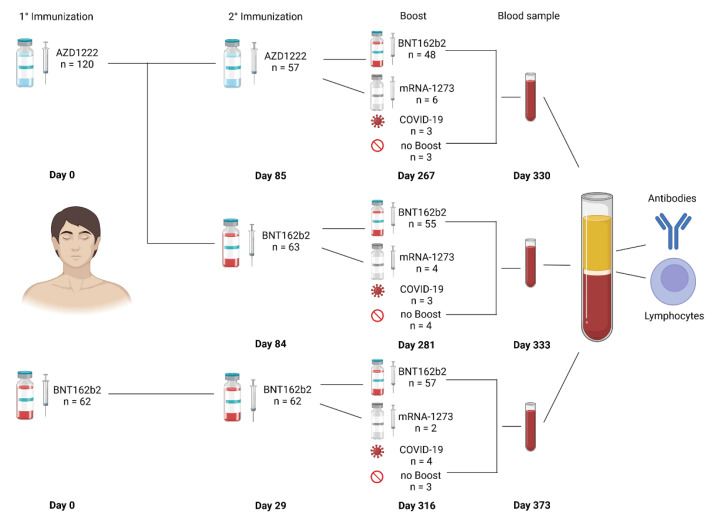
Vaccination regimens were grouped according to the respective prime/boost. Homologous and heterologous prime/boost regimen with either AZD1222/AZD1222, AZD1222/BNT162b2, or BNT162b2/BNT162b2 led to the recruitment of 182 study participants. Mean time intervals between immunizations (days) and vaccines (AZD1222 = ChAdOx1 from AstraZeneca, BNT = BNT162b2 from Pfizer/Biontech, mRNA-1273 from Moderna) are given.

**Figure 2 vaccines-10-01563-f002:**
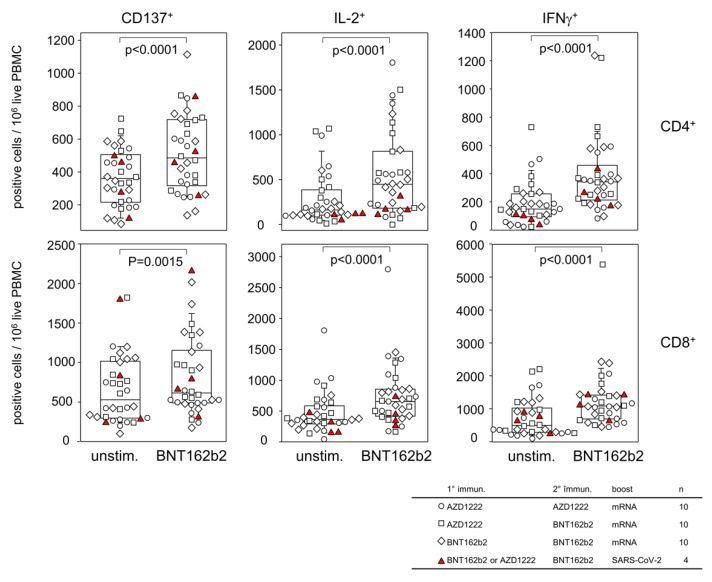
Homologous and heterologous vaccination regimen involving AZD1222 and BNT162b2 resulted in comparable T cell memories. Dot blots and corresponding box plots show increased numbers of CD137-, IL-2- and IFNγ-positive cells following in vitro re-stimulation of PBMC with BNT162b2. Different symbols represent different vaccination regimen as indicated in the table. *p* values result from Wilcoxon Matched Pairs Signed Rank Test.

**Figure 3 vaccines-10-01563-f003:**
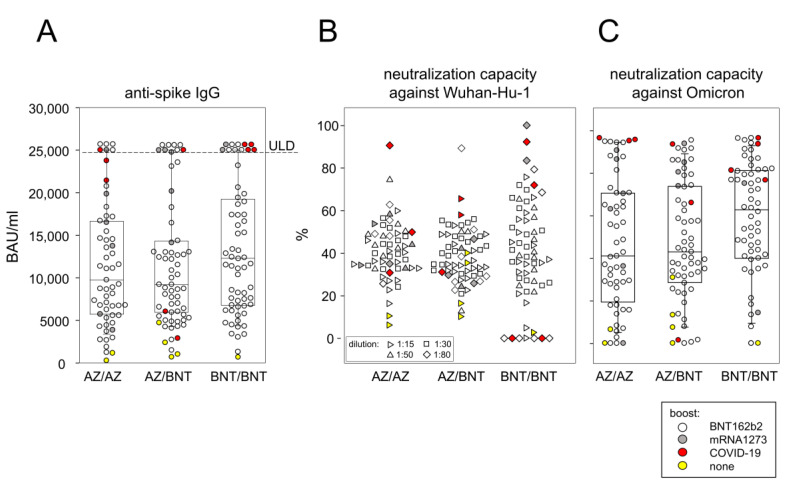
(**A**) Dot plots and corresponding box plots show anti-spike IgG concentrations (BAU/mL) in relationship to the respective vaccination regimen. The *p* value resulting from Kruskal–Wallis with post hoc test was 0.2865, indicating comparable IgG concentrations between vaccination groups. (**B**) Neutralization capacities against the Wu-Hu-1 spike protein. Serum dilutions, required to obtain neutralization capacities between 0 and 100%, are indicated by different symbols. (**C**) neutralization capacities of undiluted sera against the Omicron variant are presented in %. The *p* value resulting from Kruskal–Wallis with post hoc test was 0.0645, indicating comparable neutralization capacities between vaccination groups. Color codes indicate boost variants. AZ/AZ: homologous primary/secondary immunization with AZD1222, AZ/BNT: heterologous primary/secondary immunization with AZD1222 and BNT162b2, BNT/BNT: homologous primary/secondary immunization with BNT162b2. ULD: upper limit of detection.

**Figure 4 vaccines-10-01563-f004:**
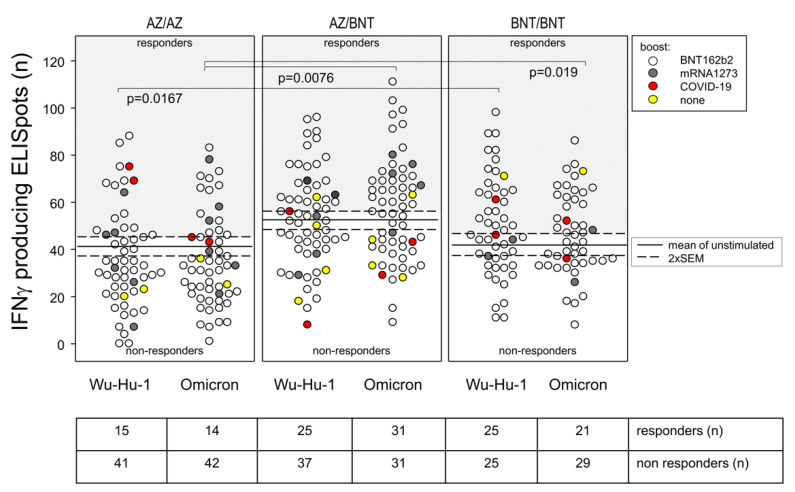
T cell memory against the Wu-Hu-1 spike protein and its Omicron variant were comparable, yet AZ-primed T cells respond less well to peptide pools. Dot plots show absolute numbers of IFNγ-producing ELISpots resulting from 5 × 10^5^ PBMCs stimulated with peptide pools representing either the wild-type spike protein (Wu-Hu-1) or its Omicron variant. Horizontal lines indicate means ± 2 SEM of IFNγ-producing ELISpots present in the absence of peptide-stimulation. Each dot represents one individual, and in the case of more than mean + 2 SEM IFNγ-producing ELISpots, individuals were considered responders. *p* values resulted from chi-squared tests, comparing the ratios of responders to non-responders among vaccination groups. AZ/AZ: homologously primed with AZD1222, AZ/BNT: heterologously primed with AZD1222 and BNT162b2, BNT/BNT: homologously primed with BNT162b2. The color code indicates boost variants.

**Figure 5 vaccines-10-01563-f005:**
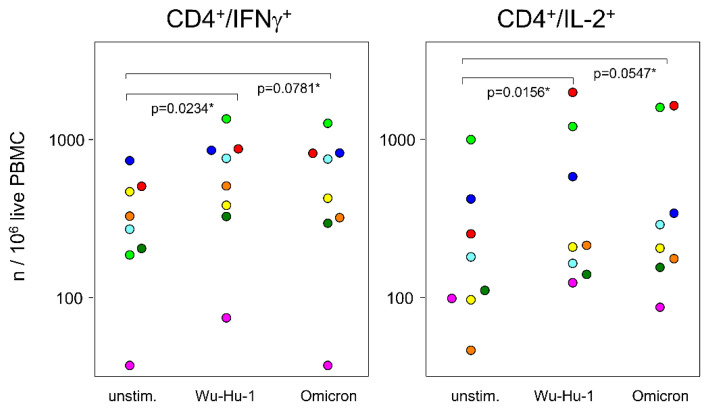
The significant drop in Omicron recognition observed for antibodies does not hold true for T cell memory. Dot plots show absolute numbers of cytokine positive T helper cells. 8 × 10^5^ PBMCs were re-stimulated with peptide pools representing either the wild-type spike protein (Wu-Hu-1) or its Omicron variant. Identical colors indicate identical vaccinees, * *p* values resulted from Wilcoxon matched-pairs signed-ranks tests.

## Data Availability

Data is contained within the article or Appendix A.

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
