# Peer review of "Vaccine-Induced T-Cell and Antibody Responses at 12 Months after Full Vaccination Differ with Respect to Omicron Recognition"

_vaccines, 2022, doi:10.3390/vaccines10091563_

Round 1

Reviewer 1 Report

Review, Mai et al., Vaccines, 2022: Vaccine-induced T- and B-cell Responses at 12 Months after Full Vaccination Differ with Respect to Omicron Recognition 

Summary

In the study by Mai et al, the group proposed to study the immune memory developed 12 months after vaccination and its efficiency against Wuhan and Omicron strains. 

They examined blood samples of 13182 health care workers 12 months after immunizations. They performed B and T cell analysis and showed that full vaccination after different immunizations didn’t affect the quantity and functionality of antibodies and T cell responses. However, they observed differences between wild type and omicron strain. 

Overall, this manuscript is well written and information is pertinent. Nevertheless, some important analysis and references are missing before publication.

Major comments

Some important works done on B and T cell responses after vaccination are missing (such as Tauzin et al, 2021; Nayrac et al, 2022; Nicolas et al, 2022, …).

Introduction

1.     Authors should introduce the link between Tfh and B cells. T follicular helper (Tfh) population, which is of particular interest, provides help for B cell maturation and development of high affinity antibody (Ab) responses in the germinal center (GC) of secondary lymphoid organs. 

Results

1.     The timing of blood collection is not clear for the reader. Authors should improve this.

2.     Figure 1, I don’t understand what stars referred to. Is it to the total number of participants or per group?

3.     Authors should mention Figure 2 in the results part.

4.     For your gating strategy, what method do you used for combining the detection of SARS-CoV-2 specific signal? If not used, an OR Boolean will be indicated.

5.     Authors should represent separately the panels of the Figure 3 especially the 3B. It is too complicated to read all information for each cohort. You should also add statistics tests on the graph.

6.     Authors cannot conclude that “neutralization of the Omicron spike variant required 15- to 80-fold more antibodies than neutralization of Wu-Hu-1” without showing IC50 assay, for example.

7.     First part of the 3.5 paragraph (lines 280 to 284) should be in the previous paragraph.

8.     Throughout the manuscript, you mentioned the B cells but you never characterized B cells only some humoral responses (such as neutralization) and the quantity of Ab. In consequence, some part should be improving such as the section 3.5.

Minor comments

1.     Line 22 add “T” between CD8+ and cells.

Author Response

Reviewer 1

Comments and Suggestions for Authors

Summary

In the study by Mai et al, the group proposed to study the immune memory developed 12 months after vaccination and its efficiency against Wuhan and Omicron strains. 

They examined blood samples of 182 health care workers 12 months after immunizations. They performed B and T cell analysis and showed that full vaccination after different immunizations didn’t affect the quantity and functionality of antibodies and T cell responses. However, they observed differences between wild type and omicron strain. 

Overall, this manuscript is well written and information is pertinent. Nevertheless, some important analysis and references are missing before publication.

Major comments

Some important works done on B and T cell responses after vaccination are missing (such as Tauzin et al, 2021; Nayrac et al, 2022; Nicolas et al, 2022, …).

Introduction

  1. Authors should introduce the link between Tfh and B cells. T follicular helper (Tfh) population, which is of particular interest, provides help for B cell maturation and development of high affinity antibody (Ab) responses in the germinal center (GC) of secondary lymphoid organs. 

We appreciate this comment and agree that this particular aspect fell a little short. However, we expanded on the issue of T cell memory and absence of correlation with antibody response in our discussion, as we felt that introducing Tfh in the introductory section would be too specific and opening a side track we did not emphasise sufficiently with our results. And even though we highly appreciate the work by Tauzin and colleagues, we felt that comparing the immune response after the first immunization and our 1-year data would not quite do justice – as Zhang and colleagues specifically expanded on the time line. We could not find any publication with Nicolas as first author in pubmed.

Results

  1. The timing of blood collection is not clear for the reader. Authors should improve this.

We tried. Instead of presenting the time period between interventions, we counted “days”, starting at day 0, the day of primary immunization, followed by the days of secondary immunization, boost and blood sampling. In fact, we also like it better now and are convinced that our experimental scheme has been improved.

  1. Figure 1, I don’t understand what stars referred to. Is it to the total number of participants or per group?

Agreed – we may have been a little over-ambitious. We meant to indicate that those who became infected did so at around the time, others were boosted. However, as nucleocapsid positive vaccines are mentioned in the text, we decided to omit the stars in Figure 1 in order to prevent irritation.

  1. Authors should mention Figure 2 in the results part.

Absolutely – we now do (please, see section 3.2).

  1. For your gating strategy, what method do you used for combining the detection of SARS-CoV-2 specific signal? If not used, an OR Boolean will be indicated.

We are not exactly sure whether we understand your question/comment correctly. We assume that you refer to the Boolean tool, the FloJo software offers. In fact, we did not use it – but instead gated on CD4 and CD8-positive cells before screening for significant differences in cytokine production (as shown in the supplemental Figure). 

  1. Authors should represent separately the panels of the Figure 3 especially the 3B. It is too complicated to read all information for each cohort. You should also add statistics tests on the graph.

Thanks for pointing this out. We now split Figure 3 into A, B and C. Statistical comparisons via Kruskal-Wallis with post hoc tests resulted in p values of 0.2865 for “A” and 0.0645 for “C”, indicating no significant differences between the various vaccination regimen. This information is now provided in the figure legend. Figure 3B served to illustrate that, in order to obtain neutralization capacities in the range of 0 – 96,5 %, serum samples needed to be diluted. Statistical comparisons of neutralization capacities between the three vaccination regimen would fail to adequately take these different dilutions into account and were therefore omitted. We do though elaborate on this in paragraph 3.4.

  1. Authors cannot conclude that “neutralization of the Omicron spike variant required 15- to 80-fold more antibodies than neutralization of Wu-Hu-1” without showing IC50 assay, for example.

Agreed, concluding from a need for an 80-fold dilution that 80-fold less antibody is required to obtain comparable neutralization capacities is not correct. We changed the last point of our summary to: “ii) neutralization of the Omicron spike variant was less efficient. “ (please, see end of section 3.4)

  1. First part of the 3.5 paragraph (lines 280 to 284) should be in the previous paragraph.

Done.

  1. Throughout the manuscript, you mentioned the B cells but you never characterized B cells only some humoral responses (such as neutralization) and the quantity of Ab. In consequence, some part should be improving such as the section 3.5.

Thanks for pointing this out. Agreed – and done. We not only changed the title of our manuscript accordingly, but also all text passages throughout the manuscript.

Minor comments

  1. Line 22 add “T” between CD8+ and cells.

Done.

Reviewer 2 Report

The paper by Mai et al. describes vaccine-induced responses of T cells and B-cells 12 months from vaccination against COVID-19. Taking into account increasing concerns about efficacy of different vaccine types and vaccination regimes, this paper confirms previously published data. Furthermore, the authors made an attempt to differentiate hybrid immunity from vaccination-induced. The paper shows data prepared in a very meticulous manner, which is of value taking into account complexity of vaccination status of the vaccinee. In my opinion, having access to PMBCs samples of vcaccinees should prompt to more detailed studies. Are the presented data the only ones, which were significant? Have the authors studied any other markers/cytokines? I think I commnet on the selected parameters to be studied by flow cytometry is necessary.

Editing remarks: why paragraphs and titels do not start from capital letters?

Author Response

Reviewer 2

Comments and Suggestions for Authors

The paper by Mai et al. describes vaccine-induced responses of T cells and B-cells 12 months from vaccination against COVID-19. Taking into account increasing concerns about efficacy of different vaccine types and vaccination regimes, this paper confirms previously published data. Furthermore, the authors made an attempt to differentiate hybrid immunity from vaccination-induced. The paper shows data prepared in a very meticulous manner, which is of value taking into account complexity of vaccination status of the vaccinee. In my opinion, having access to PMBCs samples of vcaccinees should prompt to more detailed studies. Are the presented data the only ones, which were significant? Have the authors studied any other markers/cytokines? I think I commnet on the selected parameters to be studied by flow cytometry is necessary.

Thanks for pointing this out. We introduced a table into the supplements showing the results of all Parameters analyzed- and refer to it in last results section.

Editing remarks: why paragraphs and titels do not start from capital letters?

Sorry – cannot answer that myself – but altered accordingly.

We would like to thank the reviewers for taking the time and effort to comment on our manuscript and we do hope that we addressed all issues to the full extent. We are now looking forward to a positive review.

Sincerely

Round 2

Reviewer 1 Report

Thank you for considering my comments. Great job!